# Direct synthesis of sila-benzoazoles through hydrosilylation and rearrangement cascade reaction of benzoazoles and silanes

Tianwei Liu[1,3], Mo Yang[2,3], Jianghua He[1], Shuhua Li [2] ✉ & Yuetao Zhang [1] ✉

Sila-isosteres have attracted increasing attention due to their potential application in a variety of fields and their different properties compared to their carbon-containing analogs. However, the preparation of these silicon-containing compound remains challenging and thus the development of alternative synthetic methodologies is desirable. Here, we employ $B(C_6F_5)_3$ as catalyst to enable the synthesis of highly functionalized sila-benzoazoles via hydrosilylation and rearrangement cascade reaction of benzoazoles and commercially available silanes. This strategy also exhibits remarkable features such as 100% atom-economy, good functional group tolerance, broad substrate scope, easy scale-up and good catalytic performance, demonstrating its potential application in sila-isostere synthesis.

Sila-isosteres, selectively replacing carbon with silicon within bioactive compounds, not only maintain or enhance biological activities, but also exhibit superior properties to their carbon-containing analogs, such as higher stability and lipophilicity, enhanced resistance to enzymatic degradation, increasing the membrane crossing efficiency and bioavailability, etc[1–9]. However, compared with the well-established synthesis of carbon compounds, it remains challenging to synthesize such silicon-based mimics. Despite many advancements have been achieved in recent years[1–5,10–14], direct access to the sila-isosteres by using their carbon analogs as starting materials still waits for exploration. Conventional methods for synthesis of such silicon-based mimics are typically time-consuming and inefficient, thus greatly restricting the practical application of sila-substitutions. Benzoazoles and their derivatives are widely found in biologically active molecules, natural products and fluorescent probes[15–21]. Their sila-substitution products also demonstrated special applications in sila-isostere synthesis, ligand synthesis and organic optoelectronic materials (Fig. 1a)[22–25]. However, conventional strategies for synthesis of sila-benzoazoles suffered from low atom efficiency and synthetic difficulty (vide infra, Fig. 1b and Supplementary Fig. 1). To the best of our knowledge, direct access to the sila-benzoazoles by using benzoazole derivatives as starting materials still waits for exploration. not even mentioning the highly efficient strategy with broad substrate scope, which is essentially important for high throughput screening of the bioactive compounds. Bearing these thoughts in mind, we devoted our efforts to developing synthetic strategies for sila-benzoazoles.

Here we employ a powerful and versatile Lewis acidic $B(C_6F_5)_3$ as catalyst to directly synthesize sila-benzoazoles from the reaction of benzoazole derivatives and commercially available silanes. This strategy works for a broad range of substrates, including benzothiazoles, benzoxazoles, and benzoselazoles with different substituents, as well as various types of silanes (even including a 4,4'-biphenyl-disilane), furnishing a wide variety of highly functionalized sila-benzoazoles. This work provides an example that sila-benzoazole can be prepared directly from their carbon analogs, which greatly improves the synthetic efficiency. Besides, this strategy exhibits several noteworthy features, such as excellent catalytic performance, broad substrate scope, easy scale-up, and 100% atom-economy, thus providing a simple and efficient method for direct synthesis of highly functionalized sila-isosteres from their carbon analogs (Fig. 1c). Moreover, systematical mechanistic studies involving in-situ NMR reaction, characterization of key reaction intermediates and control experiments and DFT calculations are also conducted.

[1]State Key Laboratory of Supramolecular Structure and Materials, College of Chemistry, Jilin University Changchun, Jilin 130012, China. [2]Key Laboratory of Mesoscopic Chemistry of Ministry of Education, Institute of Theoretical and Computational Chemistry, School of Chemistry and Chemical Engineering, Nanjing University, Nanjing 210093, China. [3]These authors contributed equally: Tianwei Liu, Mo Yang. ✉e-mail: shuhua@nju.edu.cn; ytzhang2009@jlu.edu.cn

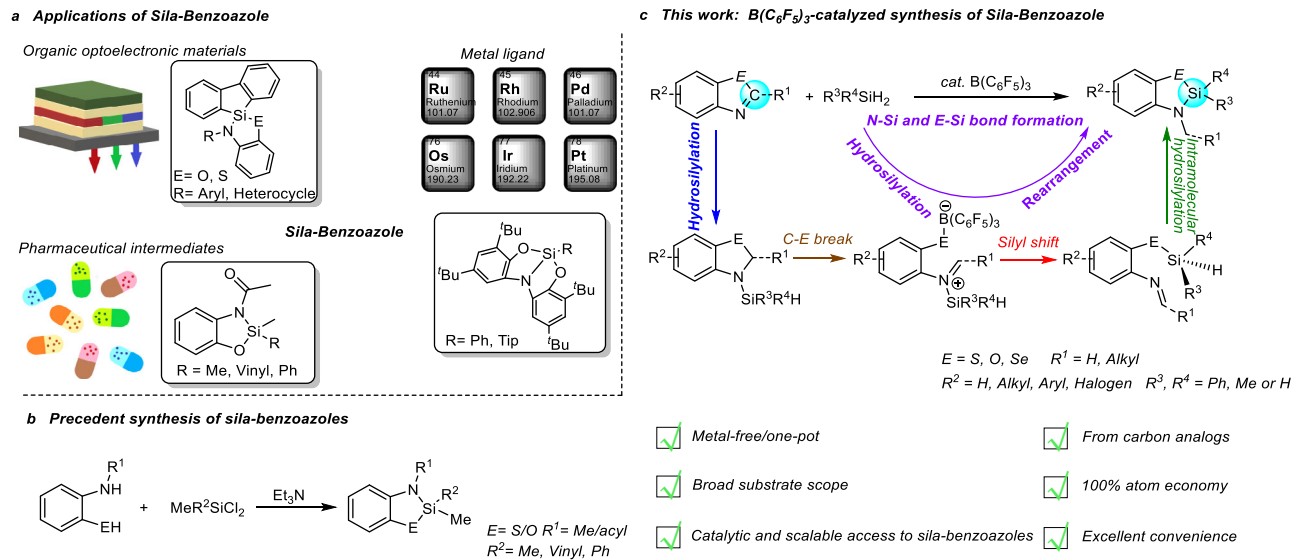

**Fig. 1 | Background of sila-benzoazoles. a** Application of sila-benzoazoles. **b** Precedent synthesis of sila-benzoazoles. **c** B(C$_6$F$_5$)$_3$-catalyzed synthesis of sila-benzoazole in this work.

## Results and discussion

### Optimization of reaction conditions

We initiated our investigation with the optimization of reaction conditions, by using benzothiazole (**1a**) and Ph$_2$SiH$_2$ (**a**) as model substrates and 5 mol% B(C$_6$F$_5$)$_3$ as catalyst. After screening reaction temperature and time (entries 1-6, Supplementary Table 1), optimal result was obtained from the reaction heating at 110 °C for 8 h and the desired product, 3-methyl-2,2-diphenyl-2,3-dihydrobenzo[d][1,3,2] thiazasilole (**6aa**) was produced in 95% yield. The solid structure of **6aa** was unambiguously confirmed by X-ray crystallographic analysis (see more details in the Supplementary Information). It is worth noting that **6aa** is the silicon-containing compound of a drug core structure that can reduce the activity of uric acid, and of a metal palladium ligand as well[18,26]. Lower reaction temperature or shorter reaction time furnished adduct **4a** and hydrosilylation intermediate 3-(diphenylsilyl) −2,3-dihydrobenzo[d]thiazole (**5aa**) rather than the desired product. We further examined the effectiveness of other Lewis acidic catalysts on reaction and found that only Et$_3$Si$^+$B(C$_6$F$_5$)$_4^-$ showed activity but much inferior to B(C$_6$F$_5$)$_3$ (entries 7–12, Supplementary Table 1). Moreover, the Ni-based catalyst capable of activating silane was also ineffective for the reaction (entry 13, Supplementary Table 1)[27]. Furthermore, our method exhibited 100% atom economy, which was in sharp contrast to traditional method starting from chlorosilane/triethylamine (Check Supplementary Fig. 1 for more details). In addition, the intermediate material o-methylaminothiophenol in the traditional method is very easily oxidized to disulfanediylbis(N-methylaniline), which undoubtedly increases the synthetic difficulty of the traditional method. As we can see above, this strategy not only enabled silicon to directly substitute C2-carbon of benzoazoles, but also retained and transferred C2-carbon onto N atom, thus realizing the direct synthesis of sila-isosteres from carbon analogs and 100% atom economy. This strategy provides a simple and convenient way to synthesize sila-substitutions, which may promote the synthesis and application of sila-benzoazoles in the future.

### Expand substrate scope

With optimized reaction conditions in hand, we next expanded the substrate scope (Fig. 2), including benzothiazole, benzoxazole, and benzoselazole (**1**−**3**). It is noted that even with 1 mol% catalyst loading, gram-scale reaction of **1a** (1.35 g) and Ph$_2$SiH$_2$ can produce **6aa** in 97% isolation yield, indicating potential application of this method in large-

scale synthesis. **1b** with electron-donating methoxy group substituted at C6-position produced **6ba** in a decreased yield of 77%, even with the prolonged reaction time. Remarkably, Cl- or Br-substituted benzothiazole (**1c**, **1d** and **1e**) all furnished corresponding sila-benzothiazole products in excellent yields (**6ca** 95%, **6da** 97%, **6ea** 99%), leaving reactive halogen atom intact, which suggested that this strategy could be employed for synthesis of sila-benzothiazoles with further functionalization capability. Considering the potential photo-electric application of sila-benzoazoles, benzothiazole with thiophene (**1f**), benzothiophene (**1g**), biphenyl (**1h**) substituted at C5-position and naphthalene (**1i**) at C6-position were utilized as substrates to synthesize corresponding products in good to excellent yields, respectively (**6fa** 95%, **6ga** 94%, **6 ha** 77%, **6ia** 80%). Furthermore, **1j** with C6-position substituted with -Bpin, a common coupling group for Suzuki reaction, afforded high yield of **6ja** (87%) retaining -Bpin group after reaction, which provided possibility for post-modification of sila-benzothiazole. It should be noted that **1j** would be mainly converted to hydrosilylation intermediates rather than **6ja** under standard condition unless high concentration of **1j** was utilized. An excellent yield of **6ka** (97%) can be also obtained when **1k** with 3,5-bis(trifluoromethyl) phenyl substituted at C6-position was utilized as substrate. However, switching to **1l** with pyridine group substituted at C5-position almost quenched the reaction (**6la**, 2%). This strategy is also applicable to benzoxazole. The reaction of benzoxazole bearing H (**2a**) or Me group (**2b** or **2c**) and Ph$_2$SiH$_2$ furnished corresponding sila-benzoxazole derivatives in high yields (**7aa** 83%, **7ba** 94%, **7ca** 92%), respectively. **2d** with electron-withdrawing chloride group at C5-position afforded the desired product **7da** in 85% yield whereas **2e** with C6-NO$_2$ substituent produced **7ea** in drastically decreased yield of 49%, probably due to the partial reduction of the nitro group in **2e** to amino group[28]. Furthermore, using benzoselazole as substrate afforded sila-benzoselazole **8aa** in 99% yield (Fig. 2). Both the preparation scale reaction and broad substrate scope clearly highlighted the practical application potential of this strategy in the future.

We further investigated the influence of substituent at C2-position of substrates on the silane insertion reaction. As shown in Fig. 3, when the C2-position of substrates, including benzothiazole, benzoxazole, and benzoselazole, was occupied by methyl group, the silane insertion reaction still proceeded smoothly. Substrates bearing electron-donating group (Me) or electron-withdrawing group (Cl, Br, Ph) on the aryl moiety all furnished the desired products in high to excellent

**Fig. 2 | Scope of benzoazoles. a** Condition: 0.1 mmol substrate, 0.1 mmol Ph$_2$SiH$_2$, 5 mol% B(C$_6$F$_5$)$_3$ in 0.6 mL CDCl$_3$, 110 °C, 8 h. Yields were measured by $^1$H NMR using mesitylene as internal standard (isolation yield in parenthesis). **b** Gram scale reaction, 10 mmol substrate, 10 mmol Ph$_2$SiH$_2$, 1 mol% B(C$_6$F$_5$)$_3$ 110 °C, 24 h. **c** 110 °C, 24 h. **d** 0.4 mmol Ph$_2$SiH$_2$. **e** 0.2 mmol **1j**, 0.2 mmol Ph$_2$SiH$_2$, 5 mol% B(C$_6$F$_5$)$_3$ in 0.2 mL CDCl$_3$, 110 °C, 8 h.

yields (**6ma** 97%, **6na** 99%, **6oa** 98%, **7fa** 98%, **7ga** 83%, **7ha** 86%, **7ia** 96%, **7ja** 94%, **8ba** 83%). This strategy can be also applied to 2-methylnaphthothiazole **1p** and 2-methylnaphthoxazole **2k**, furnishing corresponding products **6pa** and **7ka** in 92% and 96% yields, respectively. The above-obtained results prompted us to further investigate both the steric hindrance and electronic effects of C2-substituents on silane insertion reaction (Fig. 3). In comparison with 95% yield of **6aa** obtained for **1a** with a C2-H atom, increasing the bulkiness of C2 substituent would drastically decrease product yield. The bulkier substituent, the lower product yield (**1q** with C2-ethyl: **6qa** 86% vs **1r** with C2-isopropyl: **6ra** 47%; **1s** with C2-cyclopropyl: **6sa** 75% vs **1t** with C2-cyclohexyl: **6ta** 62%). The electronic properties were also found to be important for silane insertion reaction. For example, **1u** with a C2-substituted phenyl group was ineffective for this reaction, probably because in the isomerization process, the cation on N in **Int3**-like intermediate (C2-Ph) cannot be stabilized by phenyl, thus preventing its further conversion to product[29]. Interestingly, switching to C2-bromo or C2-chloro-substituted benzothiazole (**1v** and **1w**) led to the production of **6aa** other than the desired products. Increasing the amounts of Ph$_2$SiH$_2$ can significantly enhance the yield of **6aa** from 50% to 97%. It might be attributed to the reduction reaction that occurred between C2-Cl/Br of benzothiazole and Ph$_2$SiH$_2$, which furnished bromo or chloro-substituted Ph$_2$SiHCl/Br and benzothiazole **1a**, the subsequent silane insertion reaction produced **6aa** (Fig. 3)[30]. All these results demonstrated the capability of this strategy in synthesizing compounds with further derivation potentials.

Next, we expanded the scope of silanes and found that this strategy can be applied to various types of silanes (Fig. 4). When diethylsilane (Et$_2$SiH$_2$, **b**) with two ethyl groups or methylphenylsilane (PhMeSiH$_2$, **c**) with one methyl group and one phenyl group were used as silicon source, all three benzoazoles provided corresponding products in excellent yield (**6ab** 97%, **7ab** 95%, **8ab** 98%, **6ac** 98%, **7ac** 98%, **8ac** 99%). Switching to phenylsilane (**d**) with one hydrogen and one phenyl group still worked for silane insertion but exhibited different reactivities by three benzoazoles, furnishing **6ad** in 82% yield for **1a**, **7ad** in 58% yield for **2a**, and **8ad** in 84% yield for **3a**, respectively.

It is noted that the remaining Si-H bonds would provide further functionalization potentials. More recently, the stepwise polymerization based on the B(C$_6$F$_5$)$_3$-catalyzed elementary reactions of small molecules has attracted intense attention[31–36]. We also tried to examine whether this strategy can be utilized for synthesis of poly-sila-benzoazoles, a silicon analog of poly-benzoazoles showing versatile potential applications in the field of materials[37–41]. By using 4,4'-biphenyl-disilane as silicon source (**e**), benzothiazole (**1a**), 5-benzothiophenylbenzothiazole (**1g**), 2-methylnaphthothiazole (**1p**) and 2-methylnaphthoxazole (**2k**) all exhibited high reactivity and furnished the desired products in high yield (**6ae** 89%, **6ge** 73%, **6pe** 90%, **7ke** 86%). Switching to benzoselazole (**3a**) produced bis-sila-benzoselenide **8ae** in 60% yield. These results indicated that this strategy might be employed for synthesis of poly-sila-benzoazoles (Fig. 5). All the above-described results clearly demonstrated that this strategy can be applied to not only benzoazoles with different

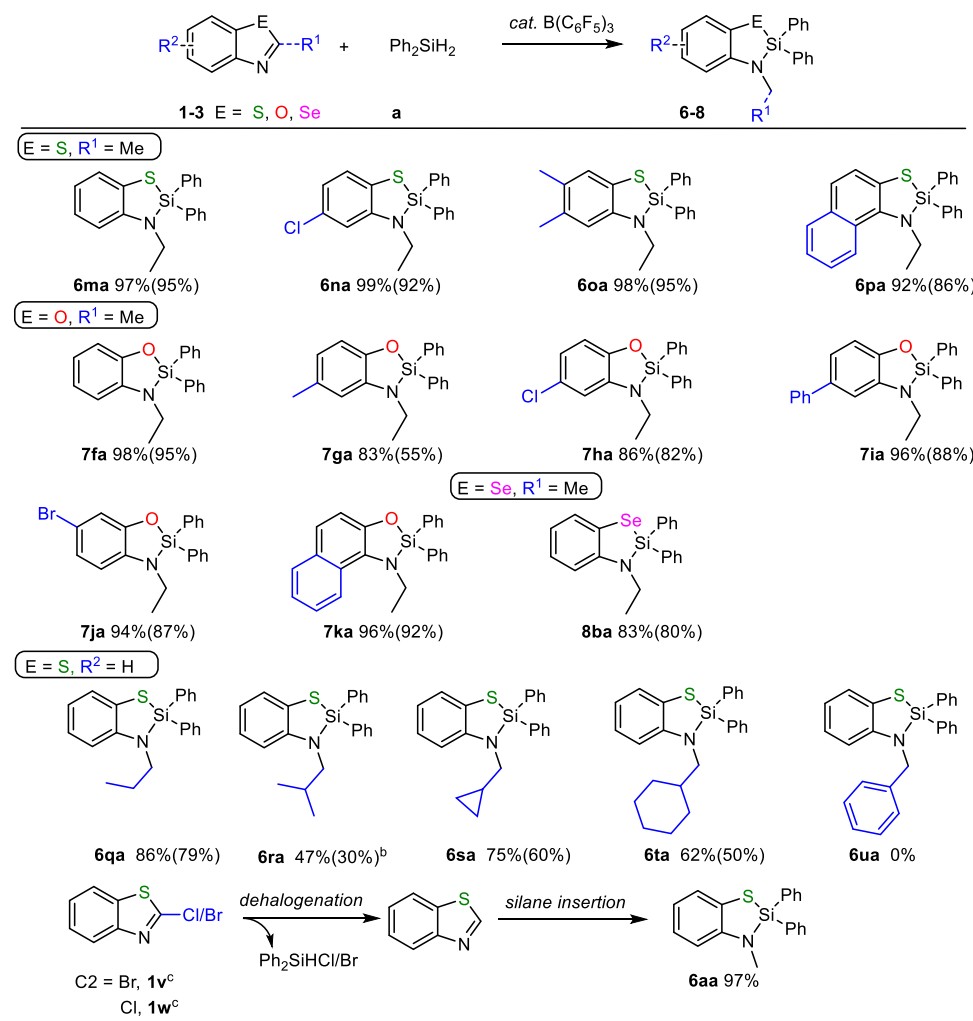

**Fig. 3 | Scope of C2-substituted benzoazoles. a** Condition: 0.1 mmol substrate, 0.1 mmol Ph₂SiH₂, 5 mol% B(C₆F₅)₃, 0.6 mL CDCl₃, 110 °C, 8 h. Yields were measured by [1]H NMR using mesitylene as internal standard (isolation yield in parenthesis). **b** 0.4 mmol Ph₂SiH₂. **c** 0.2 mmol Ph₂SiH₂.

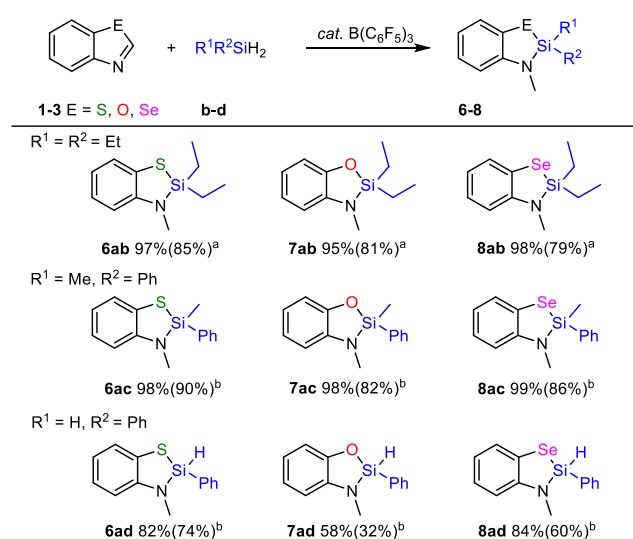

**Fig. 4 | Scope of silanes. a** Condition: 0.1 mmol substrate, 0.2 mmol Et₂SiH₂, 5 mol% B(C₆F₅)₃ in 0.6 mL CDCl₃, 110 °C, 8 h. Yields were measured by [1]H NMR using mesitylene as internal standard (isolation yield in parenthesis). **b** 0.1 mmol silane.

heteroatoms (E=S, O, Se), wide variety of electron-withdrawing/donating substituents on the aryl part or different C2-substituents of benzoazoles, but also silanes ranging from simple ones to complex ones, achieving 46 sila-benzoazoles in total and demonstrating very good generality in sila-isostere syntheses.

### Research on the reaction mechanism

To shed light on the reaction pathway, we performed a series of in-situ NMR experiments to monitor the silane insertion reaction. With consumption of **1a** (black dot), the amounts of 1,2-addition intermediate **5aa** gradually increased and then decreased (red dot). After roughly reaching 95% conversion of **1a**, the produced **5aa** started to be transformed into **6aa** (orange dot). DFT calculation (see below) revealed that the energy for the first step of hydrosilylation is lower than that for the second step of isomerization. Therefore, B(C₆F₅)₃ catalyzed the hydrosilylation rather than isomerization in the presence of large amounts of substrate **1a**. Upon reaching near quantitative conversion of **1a**, the isomerization process occurred. These results are consistent with our initial observation, in which **5aa** was generated from hydrosilylation, followed by a rearrangement reaction to furnish **6aa**. [19]F NMR spectrum only showed one set of signals attributed to the adduct generated from the reaction of **1a** and B(C₆F₅)₃, suggesting that adduct **4a** was a resting species, which was observed throughout the reaction

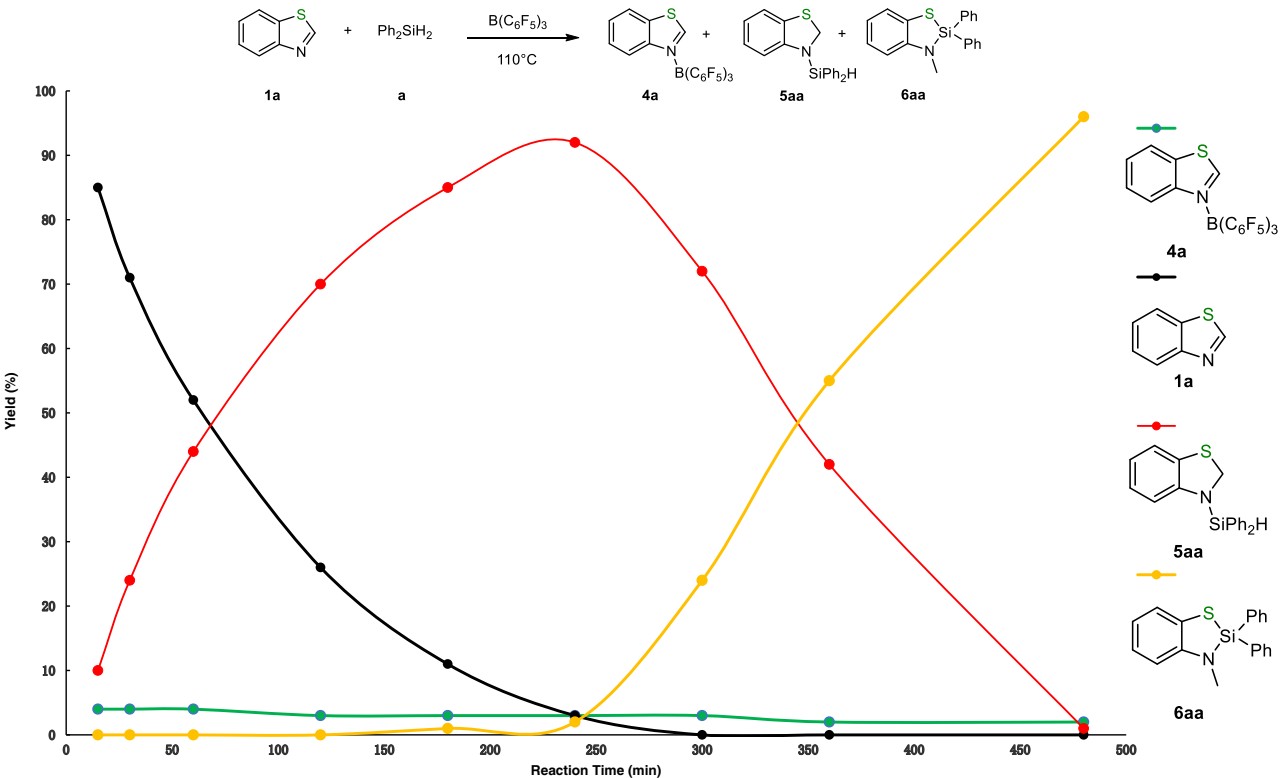

**Fig. 5 | Synthesis of Bis-sila-benzoazoles.** Condition: 0.1 mmol substrate, 0.05 mmol silane, and 5 mol% B($C_6F_5$)$_3$ in 0.6 mL $C_6D_6$, 110 °C, 8 h. Yields were measured by [1]H NMR using mesitylene as internal standard (isolation yield in parenthesis).

**Fig. 6 | Monitoring of the reaction progress.** Monitoring of the production of **6aa** from silane insertion reaction catalyzed by B($C_6F_5$)$_3$.

and confirmed by [1]H NMR spectroscopy (green dot, Fig. 6 and Supplementary Figs. 2-3). Furthermore, both [1]H and [19]F NMR spectroscopy revealed that the stoichiometric NMR reaction of **1a** and B($C_6F_5$)$_3$ generated the same species as detected in the monitoring process. [1]H NMR spectrum showed that the signal attributed to the C2-H shifted downfield when **1a** reacted with B($C_6F_5$)$_3$ in CDCl$_3$, probably due to the coordination of nitrogen with boron resulting in the decrease of electron density over the nitrogen atom. Meanwhile, [19]F NMR spectrum of **4a** showed multiple peaks in sharp contrast to that of pure B($C_6F_5$)$_3$ (Fig. 7a, Supplementary Figs. 4, 5). Control experiment

**Fig. 7 | Control experiments for the B($C_6F_5$)$_3$ catalyzed silane insertion into benzothiazole 1a.** The reaction of **a 1a** and B($C_6F_5$)$_3$; **b 1a** and Ph$_2$SiD$_2$; **c 1a-D** and Ph$_2$SiH$_2$; **d 1a** with Ph$_3$SiH or Et$_3$SiH, respectively.

showed that the reaction of deuterium-labeled **Ph$_2$SiD$_2$** with **1a** produced **6aa**, **6aa-D**, **6aa-D2**, and **6aa-D3** with deuterium partially or fully substituted the hydrogen of NCH$_3$ (Fig. 7b, Supplementary Fig. 6). The same products were obtained when C2-deuterium labeled **1a-D** was utilized to react with Ph$_2$SiH$_2$ (Fig. 7c, Supplementary Fig. 6), thus indicating that the hydrogen of NCH$_3$ in the silane insertion product came from both silane and C2-H of **1a**. Due to the abstraction of hydride by B($C_6F_5$)$_3$, the occurrence of H/D exchange of the N-CH$_3$ or reversible hydrosilylation step led to the formation of **6aa**, **6aa-D**, **6aa-D2**, and **6aa-D3** mixture[10,11]. Furthermore, control experiments revealed that tertiary silanes, such as Ph$_3$SiH and Et$_3$SiH, did not proceed with this reaction, indicating that the silane possessing at least two Si-H bonds works for this strategy (Fig. 7d, Supplementary Fig. 7).

To gain insight into the reaction mechanism, we have investigated the free energy profile of the B($C_6F_5$)$_3$–catalyzed reaction between benzothiazole **1a** and Ph$_2$SiH$_2$ with DFT calculations. The computational details are provided in the Supplementary Information. Our search for intermediates and transition states involved in the studied reaction was assisted by the molecular dynamics/coordinate driving method developed by one of the authors[42]. The results are shown in Fig. 8. First, B($C_6F_5$)$_3$ could coordinate with benzothiazole **1a** to form a stable resting species (being exothermic by 14.0 kcal mol$^{-1}$). Then, the Si-H bond of silane is activated in a concerted way with B($C_6F_5$)$_3$ as a Lewis acid and benzothiazole as a Lewis base, generating the ion–pair species **Int1**. This step is slightly exothermic by 2.7 kcal mol$^{-1}$ with respect to three isolated reactants. Then, the borohydride anion transfers a hydride to the electrophilic carbon atom (via **TS1/2**) to form hydrosilylation intermediate **5aa** (**Int2**), which is 11.1 kcal mol$^{-1}$ below the reactants. This process involves a free energy barrier of 23.6 kcal mol$^{-1}$. The calculated results are in good accord with the experimental facts that the hydrosilylation product **5aa** (**Int2**) can be obtained in 83% NMR yield under 80 °C for 8 h (entry 4 in Supplementary Table 1)[43]. In the hydrosilylation step, since B($C_6F_5$)$_3$ has the ability to abstract hydride, transformation from **Int1** to **Int2** through the hydride transfer from the borohydride anion to the electrophilic carbon atom is reversible, which is consistent with D-scrambling observed in the experiments of Figs. 7b and 7c and DFT results[11,44]. The concerted

activation of the Si-H bond in Ph$_2$SiH$_2$ by B($C_6F_5$)$_3$ and **1a** is similar to that in hydrosilylation/hydroboration reactions catalyzed by B($C_6F_5$)$_3$ reported previously[45–50].

In the next stage, the addition of B($C_6F_5$)$_3$ onto the nucleophilic S atom of **5aa** (**Int2**) would induce the breaking of the C-S bond (via **TS2/3**) to give a ring-opening species **Int3**. This step involves a barrier of 24.0 kcal mol$^{-1}$ (relative to **Int2**). Then, species **Int3** is likely to isomerize to a more stable species **Int4** through a C-N bond rotation. The nitrogen-bound silyl group subsequently shifts to the S atom through electrophilic attack to give **Int5** (via **TS4/5**), releasing B($C_6F_5$)$_3$. Then, B($C_6F_5$)$_3$ approaches the remaining hydride on the silyl group to form **Int6**. The hydride abstraction by B($C_6F_5$)$_3$ generates an ion-pair intermediate **Int7** (via **TS6/7**). In the final step, a subsequent hydride transfer from boron to N-methylene would readily occur, forming the final product **6aa** and regenerating the catalyst B($C_6F_5$)$_3$. The overall reaction of the second stage (from **Int2** to **6aa**) is exothermic by 26.0 kcal mol$^{-1}$, with a barrier of 30.9 kcal mol$^{-1}$ in the rate-limiting step (the silyl shift step). The catalytic cycle of this stage undergoes a ring isomerization process, in which both a silyl shift and hydride transfer are involved. The calculated results can also account for the deuterium-labeling results and the experimental facts that a higher temperature (110 °C) was required (entry 6 in Supplementary Table 1 and Fig. 9).

To provide more experimental evidence to support the ring isomerization process, we tried to investigate the reaction process through kinetic studies but failed, possibly due to the high reaction rate. Attempts to synthesize **Int5** also failed. Fortunately, by replacing one H-atom of N-methylene with a methyl group, we successfully prepared **Int5-Me**, which could completely be transformed into **6ma** with the catalysis of B($C_6F_5$)$_3$ (Supplementary Fig. 10). This result is consistent with the calculated ring isomerization process. All the above-mentioned experimental and calculated results provided strong evidence to support the proposed cascade hydrosilylation/ring-isomerization mechanism, which was summarized in Fig. 9.

In summary, we developed a simple method to access up to 46 highly functionalized sila-benzoazoles with C2-carbon atom directly substituted by silanes in high to excellent yields, furnishing the direct

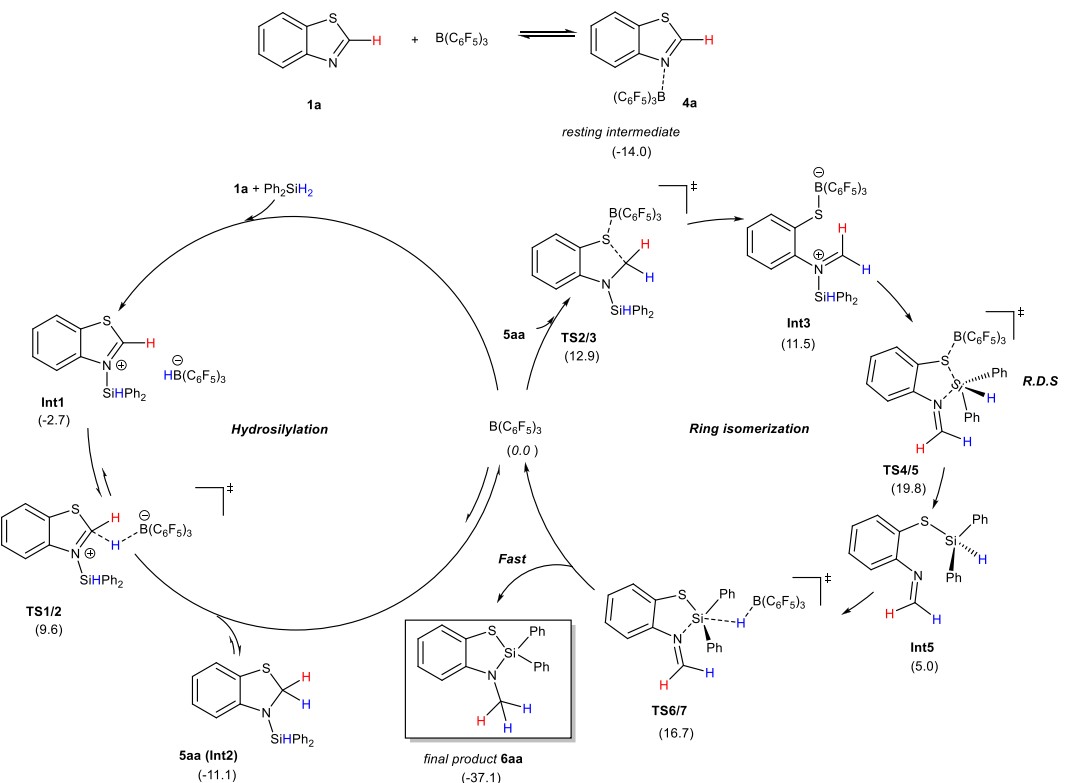

**Fig. 8 | DFT calculations on the proposed reaction pathway.** Gibbs free energy profile for the B(C₆F₅)₃ catalyzed silane insertion into benzothiazole **1a** (in kcal mol⁻¹).

**Fig. 9 | Reaction mechanisms.** The mechanism proposed for the B(C₆F₅)₃-catalyzed synthesis of sila-benzoazole (in kcal mol⁻¹).

synthesis of sila-isosteres from their carbon analogs. This strategy showed broad substrate scope, including 23 benzothiazoles, 11 benzoxazoles and 2 benzoselazoles and 5 silanes, even a di-linker 4, 4′-biphenyl-disilane can be utilized for synthesis of poly(sila-benzoazoles).

Through systematic mechanistic investigations including in-situ NMR reaction, identification of key reaction intermediates and control experiments coupled with DFT calculations, we successfully elucidated that such sila-insertion reaction proceeded through cascade

hydrosilylation and ring isomerization. Furthermore, its good functional group tolerance, easy scale-up and 100% atom economy demonstrated the powerful capability of such a strategy in synthesizing the silicon-based mimics to their carbon analogs. This strategy showed very promising prospects in practical applications, which may benefit the development of drug candidates.

## Methods

### General procedure for the reaction

In a glovebox, benzoazole and $B(C_6F_5)_3$ were dissolved in $CDCl_3/C_6D_6$ (0.6 mL) in a 2-mL NMR tube, then corresponding silane and mesitylene were added to the above mixture. The reaction was heated at 110 °C for specific time, then measured by $^1H$ NMR spectroscopy. After the reaction, the reaction solvent was removed under vacuum to obtain a mixed product, the mixture was washed with anhydrous hexane or vacuum distilled to obtain the final product.

### Gram-scale reaction

In a glovebox, **1a** (1.35 g, 10 mmol) and $B(C_6F_5)_3$ (51.2 mg, 0.1 mmol) were dissolved in $CHCl_3$ (2 mL) in a 20 mL pressure tube, then added with silane (1.84 g, 10 mmol). After heating the reaction at 110 °C for 24 h, a small aliquot was taken for $^1H$ NMR measurement. The reaction mixture was concentrated under vacuum and slowly added with dry hexane (5 mL × 3) and then removed the up-layer of hexane, dried in vacuo to afford the final product (97% isolation yield).

See Supplementary Information for more details.

## Data availability

All the characterization data and experimental protocols are provided in this article and its Supplementary Information. Supplementary Data 1 contains the calculated energies and Cartesian Coordinates of the optimized structures. The X-ray crystallographic data for compound **6aa**, $C_{19}H_{17}NSSi$, have been deposited at the Cambridge Crystallographic Data Center (CCDC), under deposition number 2167471. These data can be obtained free of charge from CCDC via www.ccdc.cam.ac.uk/data_request/cif.

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

## Acknowledgements

This work was supported by the National Natural Science Foundation of China (Grant nos. 22225104, 21871107, 21975102, to Y.Z. and 22073043 to S.L.). All theoretical calculations were performed at the High-Performance Computing Center (HPCC) of Nanjing University.

## Author contributions

T.L. designed and conducted experiments. M.Y. performed the DFT calculations. J.H., S.L. and Y.Z. analyzed the data and wrote the manuscript with input from all other authors.

## Competing interests

The authors declare no competing interests.
