## [Peer Review File · Nature Communications]

Direct Synthesis of Sila-Benzoazoles through Hydrosilylation and Rearrangement Cascade Reaction of Benzoazoles and SilanesREVIEWER COMMENTS

Reviewer #1 (Remarks to the Author):

According to the manuscript titled "Direct Synthesis of Sila-Benzoazoles through Hydrosilylation and Rearrangement Cascade Reaction of Benzoazoles and Silanes", the author successfully developed a method to synthesize sila-benzoazoles through hydrosilylation using metal-free $B(C_6F_5)_3$ as catalyst. The substrate scope was also broad to benzothiazoles, benzoxazoles, and benzoselazoles, which all produced reasonable to high yields depending on the substituents. Additionally, silanes such Et_2SiH_2 , $MePhSiH_2$, $PhSiH_3$ were evaluated and provided good yields. Especially, bis-sila-benzoazoles were formed when using 4,4'-biphenyl-disilane. This procedure was more efficient than the traditional one for atom-economic synthesis of sila-benzoazoles. The authors also conducted several control experiments to demonstrate the reaction pathway via the 1,2-hydrosilylation intermediate to 95% conversion, and the following transformation of the intermediate into the desired silacyclic product. Furthermore, the author has provided a plausible mechanism using deuterium-labeled Ph_2SiD_2 and deuterium(C_2)-labeled benzoazoles in conjunction with DFT calculations to show the energies in each step. I suggest that this work can be published in Nature Communications due to the novel sila-benzoazole synthesis, atom-economic procedure, excellent yields, and fully and clearly presented data after minor revisions below.

- In this manuscript, the authors wrote "chloride group at C5-position afforded the desired product 7da in 85% yield while 2e with C6-NO₂ substituent produced 7ea in severely lower yield of 49%". However, according to the following literature, "Porwal, D. and Oestreich, M. Eur. J. Org. Chem. 2016.3307-3309," the corresponding amines might be formed via $B(C_6F_5)_3$ -catalyzed hydrosilylation of aromatic and aliphatic nitro groups. Have the authors noticed any reduced amine in the reaction mixture? If this is the case, the resulting lower yield could be related to this side reaction.

- The ¹³C NMR of some products does not have a sufficient number of carbons. Could you add more detail about the number of carbon if one signal contains more than two carbon atoms? (e.g. 120.4 (2C))

Also, since your products contain Si atom, I think the authors should add ²⁹Si NMR of the resulting products in SI. These could be a good reference of the ²⁹Si NMR peak for the sila-benzoazoles.

- We noticed that the reaction for the synthesis of 6ja was highly concentrated but the authors didn't explain about the reason. The reason should be rationalized by comparing the yield of the 6ja with the normal condition.

- Reorganization / section separation of the Scheme 2 might be required for better visibility. (for E=S/O/Se)

- About the reaction of 6ua, which step of the cascade reaction was affected by the phenyl substituent? If the first hydrosilylation was not occurred, the additional stability from the imine conjugation might be related to the shut-down of the reaction. This should be discussed in the manuscript.

- Besides the interesting results and discussions in the Schemes and Tables, there are so many unclear sentences due to the English errors. I strongly recommend the authors to submit this manuscript for English correction to any specialist for clarity.

- For Scheme 5, have the authors tried the bis-benzoazole for the actual polymer synthesis? If they have, how was the result?

- In the Fig 1, Why the product 6aa was not formed during the initial 200 min? I think that the adduct 4a formation of the BCF catalyst and 1a might be related to the catalyst activity. Could you explain it in the manuscript?

- The authors mentioned that the mixture formation of the 6aa, 6aa-D, 6aa-D2, and 6aa-D3 was related to the fact that the only silanes possessing at least two Si-H bonds works for this reaction. Could you explain why?

Reviewer #2 (Remarks to the Author):

In this work, Zhang and Li groups described a direct synthetic route towards sila-benzoazoles from their parent benzoazoles using $B(C_6F_5)_3$ as a sole catalyst and secondary hydrosilanes as both hydride and silicon sources. The borane was capable of not only activating a Si-H bond, but also forming an adduct with substrate. It has been unprecedented to synthesize the sila-benzoazoles from their carbon analogs, although the sila-benzoazoles have been regarded as a class of interesting sila-isosteres in the field of medicinal, materials, and organometallic chemistry.

Given this background, the authors discovered a new reductive cascade route towards the sila-benzoazoles under heating conditions of BCF and secondary hydrosilanes. The scope of benzoazoles (starting materials) and hydrosilanes were broad in good to high yields, while this strategy was applicable for the synthesis of bis-sila-benzoazoles with high efficiency by simply employing a suitable disilane.

The catalytic pathway for this cascade reaction was elucidated through a series of preliminary mechanistic studies containing a 1H NMR monitoring (time-conversion), deuterium-labeling, and a reaction of a model intermediate. DFT calculations suggested the reasonable activation energy barriers for hydrosilylation and silyl-transfer steps, and thus the rate-determining step.

Overall, the discovered reaction pathway is new and novel and unique to provide a broad range of sila-benzoazoles with excellent convenience and atom-efficiency, which have been unattainable through the conventional stoichiometric reaction protocols. The first step of hydrosilylation is well known reactivity, whereas the second step – silyl group transfer following the E-C (E = O, S, Se) bond cleavage by an action of BCF as a Lewis acid, is a certainly new reactivity to bring out the present chemistry.

Because of the above reasons, this reviewer recommends minor revision for publication in Nature Communications. The below is a list of issues that the authors need to address prior to publication.

- (1) Overall, the English-wording including grammars needs to be polished by English experts. Some sentences are unclear due to poor description and/or lack of Scheme.
- (2) In page 1, line 18, "Sila-substitution" is better to change to "Sila-isosteres" as the biological factor.
- (3) In page 1, line 22-23, .."there are no strategy to access the silicon-based compounds directly from their carbon analogs as a starting materials yet." may be better? Needs to be revised anyway in a more concise way.
- (4) In page 1, line 23-24, ..."Typically, a new method ...sila-substitutions." needs to be rewritten using a word "conventional method".
- (5) In page 2, line 27, "organic transformation" seems overlap the "drug modification". Better to delete the organic transformation, instead "ligand synthesis" should be included. "drug modification" is not a suitable word... "sila-isostere synthesis" may be better.
- (6) In page 2, line 28, "current strategies" to "conventional strategies". ..."suffered from low atom efficiency and synthetic difficulty" is better.
- (7) In page 2, line 29, a sentence "So far, we...." Is better to rewritten to emphasize the most novel point of this development. The current version seems too weak to underline the most selling-point of this work.
- (8) In page 2, line 36, ..."a bulky 4,4'-biphenyl-disilane,..." "bulky" needs to be deleted. This molecule is not bulky. Et_3SiH or iPr_3SiH or tBu_2MeSiH are regarded as a bulky hydrosilane in reduction chemistry.
- (9) In page 2, line 37-38, better to delete "...also the first example of sila-substitution...".
- (10) In page 2, line 41-43, "systematically... containing in-situ NMR reaction, characterization of key

... are conducted". may be better?

(11) In page 3, Scheme 1, "a" needs to show the specific family of structures responsible for optoelectronic materials, metal ligand, and drug intermediates. And, better to delete the "Organic synthesis intermediates", which overlap drug intermediates and quite vague.

(12) In page 3, "b" the subtitle is better to be "Precedent synthesis of sila-benzoazoles" or "Typical reaction towards sila-benzoazoles". "c" the step "hydride transfer" may be better to change to "intramolecular hydrosilylation". In a list of pros of this catalysis, "Excellent convenience" is better to be included. "Catalytic and scalable access to sila-benzoazoles".

(13) In page 3, line 53, "a drug host structure" seems not suitable. "a drug core structure" better?

(14) In page 3, line 55, "hydrosilylation intermediate" is better.

(15) In page 4, line 67-70, the sentences "For a starting... In addition, the ... the traditional method." are certainly unclear and this reviewer is doubtful if there is need to mention the specific numbers of atom-economy for each reaction route. It is already obvious that the conventional stoichiometric reaction generates copious salts waste.

(16) "disulfanediybis(N-methylaniline)" is difficult to imagine without any structure.

(17) In page 4, line 72 – (page 5) 73, This reviewer does not like to place a statement regarding the potential of this catalysis since that is not an observation, but just a possibility or perspective. "This strategy can be a big breakthrough in synthesis of And would significantly promote...." Better to tone down in this sentence.

(18) In Scheme 3, for 6ra, what is the remaining? Did the authors observe the simply reduced product or any other side-product in 6ra or some substrates with low yields? It is better to mention regarding the general side-products observed in the crude mixture, so that authors understand which step is quite challenging to achieve this cascade reaction in good yield.

(19) In page 7, line 112, the electronic factor which hampered the reaction of 1u, needs a suitable citation.

(20) In page 7, line 116, a reference is needed for the dehalogenative reduction under the BCF-hydrosilane catalytic system.

(21) In page 11, line 188, after the sentence "... (entry 4 in Supplementary Table 1 ...)4." The authors need to discuss about the reversible formation of Int 1 from Int 2, together with a relation to the D-scrambling observed in the experiments of Scheme 6 b and 6c. In this reversible formation of Int 1, BCF works as a Lewis acid to abstract the hydride. A suitable paper(s) "DOI: 10.1039/c9qo01437c" and "Ref. 11" need to be cited for this reversibility.

(22) Accordingly, in Scheme 7, the arrows are needed between Int1 and Int2 to reflect the reversible formation of Int1 from Int2.

(23) In Scheme 7, "RDS" better to be put on the conversion flow from 5aa (via TS2/3) to Int5 in the Ring Isomerization cycle. In addition, "fast" is good to put on the arrow over from Int5 to TS6/7 to 6aa.

(24) This reviewer suggests to put the ground energies of key species containing "resting intermediate (-14.0)", BCF (0.0 kcal/mol), Int1 (-2.7), Int2 (-11.1), and product (-37.1).

Responses to the Comments by the reviewers

Reviewer 1:

Reviewer's general comments: According to the manuscript titled "Direct Synthesis of Sila-Benzoazoles through Hydrosilylation and Rearrangement Cascade Reaction of Benzoazoles and Silanes", the author successfully developed a method to synthesize sila-benzoazoles through hydrosilylation using metal-free $B(C_6F_5)_3$ as catalyst. The substrate scope was also broad to benzothiazoles, benzoxazoles, and benzoselazoles, which all produced reasonable to high yields depending on the substituents. Additionally, silanes such Et_2SiH_2 , $MePhSiH_2$, $PhSiH_3$ were evaluated and provided good yields. Especially, bis-sila-benzoazoles were formed when using 4,4'-biphenyl-disilane. This procedure was more efficient than the traditional one for atom-economic synthesis of sila-benzoazoles. The authors also conducted several control experiments to demonstrate the reaction pathway via the 1,2-hydrosilylation intermediate to 95% conversion, and the following transformation of the intermediate into the desired silacyclic product. Furthermore, the author has provided a plausible mechanism using deuterium-labeled Ph_2SiD_2 and deuterium(C2)-labeled benzoazoles in conjunction with DFT calculations to show the energies in each step. I suggest that this work can be published in Nature Communications due to the novel sila-benzoazole synthesis, atom-economic procedure, excellent yields, and fully and clearly presented data after minor revisions below.

Our response: We greatly appreciate the reviewer's positive recommendation and carefully revised the manuscript and supporting information according to the reviewer's valuable suggestion.

Reviewer's specific questions:

1. In this manuscript, the authors wrote "chloride group at C5-position afforded the desired product 7da in 85% yield while 2e with C6-NO₂ substituent produced 7ea in severely lower yield of 49%". However, according to the following literature, "Porwal, D. and Oestreich, M. Eur. J. Org. Chem. 2016.3307-3309," the corresponding amines might be formed via $B(C_6F_5)_3$ -catalyzed hydrosilylation of aromatic and aliphatic nitro groups. Have the authors noticed any reduced amine in the reaction mixture? If this is the case, the resulting lower yield could be related to this side reaction.

Our response: We greatly appreciate the reviewer's valuable comments. The in-situ NMR spectrum of this reaction revealed the presence of signals attributed to small amounts of substrate residues and some relatively chaotic peaks in the aryl region probably associated with the reduced amines. Corresponding discussion and literatures were added to the revised manuscript.

2. The ¹³C NMR of some products does not have a sufficient number of carbons. Could you add more detail about the number of carbon if one signal contains more than two carbon atoms? (e.g. 120.4 (2C)). Also, since your products contain Si atom, I think the authors should add ²⁹Si NMR of the resulting products in SI. These could be a good reference of the ²⁹Si NMR peak for the sila-benzoazoles.

Our response: According to the reviewer's valuable suggestion, we have added the number of carbon atoms into the ¹³C NMR data in the revised supplementary information. Moreover, the ²⁹Si NMR spectra for the sila-benzoazoles were also added to the revised supporting information.

3. We noticed that the reaction for the synthesis of 6ja was highly concentrated but the authors didn't explain about the reason. The reason should be rationalized by comparing the yield of the 6ja with the normal condition.

Our response: It is noted that **1j** was mainly converted into hydrosilylation intermediates under standard condition. After optimization of conditions, we found **1j** can be converted to product **6ja** under high concentration. Corresponding discussion was included into the revised manuscript.

4. Reorganization / section separation of the Scheme 2 might be required for better visibility. (for E=S/O/Se)

Our response: Scheme 2 was revised as instructed.

5. About the reaction of 6ua, which step of the cascade reaction was affected by the phenyl substituent? If the first hydrosilylation was not occurred, the additional stability from the imine conjugation might be related to the shut-down of the reaction. This should be discussed in the manuscript.

Our response: Using **1u** with a C2-substituted phenyl group was ineffective for this reaction, probably because in the isomerization process, the cation on N in **Int3**-like intermediate (C2-Ph) cannot be stabilized by phenyl, thus preventing its further conversion to product. Corresponding discussion was included into the revised manuscript.

Int3 (C2-Ph)

6. Besides the interesting results and discussions in the Schemes and Tables, there are so many unclear sentences due to the English errors. I strongly recommend the authors to submit this manuscript for English correction to any specialist for clarity.

Our response: As suggested, we had a colleague review our manuscript and also tried to improve it by ourself as much as we can.

7. For Scheme 5, have the authors tried the bis-benzoazole for the actual polymer synthesis? If they have, how was the result?

Our response: It was not applied to the polymer synthesis yet.

8. In the Fig 1, Why the product 6aa was not formed during the initial 200 min? I think that the adduct 4a formation of the BCF catalyst and 1a might be related to the catalyst activity. Could you explain it in the manuscript?

Our response: According to DFT calculation (Fig. 2), as the energy of the first step of hydrosilylation is lower than that of the second step of isomerization, the BCF catalyst will not catalyze the isomerization process but hydrosilylation until the substrate reaching near quantitative completion. Corresponding discussion was added to the revised manuscript.

9. The authors mentioned that the mixture formation of the 6aa, 6aa-D, 6aa-D2, and 6aa-D3 was related to the fact that the only silanes possessing at least two Si-H bonds works for this reaction. Could you explain why?

Our response: Sorry for the misunderstanding resulted from our unclear description. Actually, control experiment shown in scheme 6d that using tertiary silanes, such as Ph₃SiH or Et₃SiH, as substrates, did not proceed this reaction, thus indicating that the silane possessing at least two Si-H bonds works for this strategy. We corrected this sentence in the revised manuscript.

Reviewer 2

Reviewer's general comments: In this work, Zhang and Li groups described a direct synthetic route towards sila-benzoazoles from their parent benzoazoles using B(C₆F₅)₃ as a sole catalyst and secondary hydrosilanes as both hydride and silicon sources. The borane was capable of not only activating a Si-H bond, but also forming an adduct with substrate. It has been unprecedented to synthesize the sila-benzoazoles from their carbon analogs, although the

sila-benzoazoles have been regarded as a class of interesting sila-isosteres in the field of medicinal, materials, and organometallic chemistry. Given this background, the authors discovered a new reductive cascade route towards the sila-benzoazoles under heating conditions of BCF and secondary hydrosilanes. The scope of benzoazoles (starting materials) and hydrosilanes were broad in good to high yields, while this strategy was applicable for the synthesis of bis-sila-benzoazoles with high efficiency by simply employing a suitable disilane. The catalytic pathway for this cascade reaction was elucidated through a series of preliminary mechanistic studies containing a ¹H NMR monitoring (time-conversion), deuterium-labeling, and a reaction of a model intermediate. DFT calculations suggested the reasonable activation energy barriers for hydrosilylation and silyl-transfer steps, and thus the rate-determining step. Overall, the discovered reaction pathway is new and novel and unique to provide a broad range of sila-benzoazoles with excellent convenience and atom-efficiency, which have been unattainable through the conventional stoichiometric reaction protocols. The first step of hydrosilylation is well known reactivity, whereas the second step – silyl group transfer following the E-C (E = O, S, Se) bond cleavage by an action of BCF as a Lewis acid, is a certainly new reactivity to bring out the present chemistry. Because of the above reasons, this reviewer recommends minor revision for publication in Nature Communications. The below is a list of issues that the authors need to address prior to publication.

Our response: We greatly appreciate the reviewer's kind comments and carefully revised the manuscript and supporting information accordingly.

Reviewer's specific questions:

1. Overall, the English-wording including grammars needs to be polished by English experts. Some sentences are unclear due to poor description and/or lack of Scheme.

Our response: We greatly appreciate the reviewer's valuable suggestion and have read through the manuscript to try to improve it as much as we can.

2. In page 1, line 18, "Sila-substitution" is better to change to "Sila-isosteres" as the biological factor.

Our response: Changed as suggested.

3. In page 1, line 22-23, .."there are no strategy to access the silicon-based compounds directly from their carbon analogs as a starting materials yet." may be better? Needs to be revised anyway in a more concise way.

Our response: Changed according to the reviewer's valuable suggestion.

4. In page 1, line 23-24, ..."Typically, a new method ...sila-substitutions." needs to be rewritten using a word "conventional method".

Our response: Changed as suggested.

5. In page 2, line 27, "organic transformation" seems overlap the "drug modification". Better to delete the organic transformation, instead "ligand synthesis" should be included. "drug modification" is not a suitable word... "sila-isostere synthesis" may be better.

Our response: Changed as suggested.

6. In page 2, line 28, "current strategies" to "conventional strategies". ..."suffered from low atom efficiency and synthetic difficulty" is better.

Our response: Changed as suggested.

7. In page 2, line 29, a sentence "So far, we..." Is better to rewritten to emphasize the most novel point of this development. The current version seems too weak to underline the most selling-point of this work.

Our response: Changed as suggested.

8. In page 2, line 36, ...”a bulky 4,4’-biphenyl-disilane,...” “bulky” needs to be deleted. This molecule is not bulky. Et₃SiH or ^tPr₃SiH or ^tBu₂MeSiH are regarded as a bulky hydrosilane in reduction chemistry.

Our response: Changed as suggested.

9. In page 2, line 37-38, better to delete “...also the first example of sila-substitution...”.

Our response: Changed as suggested.

10. In page 2, line 41-43, “systematically... containing in-situ NMR reaction, characterization of key ... are conducted”. may be better?

Our response: Changed as suggested.

11. In page 3, Scheme 1, “a” needs to show the specific family of structures responsible for optoelectronic materials, metal ligand, and drug intermediates. And, better to delete the “Organic synthesis intermediates”, which overlap drug intermediates and quite vague.

Our response: Changed as suggested.

12. In page 3, “b” the subtitle is better to be “Precedent synthesis of sila-benzoazoles” or “Typical reaction towards sila-benzoazoles”. “c” the step “hydride transfer” may be better to change to “intramolecular hydrosilylation”. In a list of pros of this catalysis, “Excellent convenience” is better to be included. “Catalytic and scalable access to sila-benzoazoles”.

Our response: Changed as suggested.

13. In page 3, line 53, “a drug host structure” seems not suitable. “a drug core structure” better?

Our response: Changed as suggested.

14. In page 3, line 55, “hydrosilylation intermediate” is better.

Our response: Changed as suggested.

15. In page 4, line 67-70, the sentences “For a starting... In addition, the ... the traditional method.” are certainly unclear and this reviewer is doubtful if there is need to mention the specific numbers of atom-economy for each reaction route. It is already obvious that the conventional stoichiometric reaction generates copious salts waste.

Our response: We tried to make comparison between our method and the traditional method starting from chlorosilane/triethylamine and found both methods have similar product yield but different atom efficiency. More specifically, if starting from o-methylthiophenol, our method is 100% in comparison with 54% exhibited by traditional one. If starting from o-thiophenol, ours is 79% vs 35% by traditional one (See below for calculation). These results demonstrated the high atom efficiency of our method. We do agree with the reviewer that the result is so obviously. Therefore, we simplified the comparison in the main text and moved the detailed calculation results to the supplementary information.

calculate from 2-mercaptoaniline

Our pathway

$$\text{atom efficiency (Step 1)} = \frac{135.2}{125.2+30+16} \times 100\% = 79\%$$

$$\text{atom efficiency (Step 2)} = 100\%$$

$$\text{atom efficiency (finally)} = 79\% \times 100\% = 79\%$$

Traditional pathway

$$\text{atom efficiency (Step 1)} = \frac{139}{125.2+90} \times 100\% = 65\%$$

$$\text{atom efficiency (Step 2)} = \frac{319}{139+101.2 \times 2+253} \times 100\% = 54\%$$

$$\text{atom efficiency (finally)} = 65\% \times 54\% = 35\%$$

(It is worth noting that o-methylaminothiophenol will deteriorate in the presence of oxygen)

disulfanediybis(N-methylaniline)

Supplementary Scheme 1 Comparison of the atomic efficiency of our strategy with that of traditional method (taking **6aa** as example).

16. “disulfanediybis(N-methylaniline)” is difficult to imagine without any structure.

Our response: The structure of disulfanediybis(N-methylaniline) was included into the revised support information (See Supplementary Scheme 1).

17. In page 4, line 72 – (page 5) 73, This reviewer does not like to place a statement regarding the potential of this catalysis since that is not an observation, but just a possibility or perspective. “This strategy can be a big breakthrough in synthesis of And would significantly promote....” Better to tone down in this sentence.

Our response: Changed as suggested.

18. In Scheme 3, for **6ra**, what is the remaining? Did the authors observe the simply reduced product or any other side-product in **6ra** or some substrates with low yields? It is better to mention regarding the general side-products observed in the crude mixture, so that authors understand which step is quite challenging to achieve this cascade reaction in good yield.

Our response: In situ NMR spectrum obtained after reaction revealed the presence of product, substrate residue and small amounts of unidentified by-product mixtures. For reactions obtaining low product yield, prolonged reaction time did not improve the product yield, but led to the formation of even messy by-product mixtures. Taking **1r** as example, its low product yield is probably due to the low cascade reaction activity resulted from its large steric hindrance. Corresponding discussion was added to the revised manuscript.

19. In page 7, line 112, the electronic factor which hampered the reaction of **1u**, needs a suitable citation.

Our response: Reference was added to the citation list according the reviewer's valuable suggestion.

20. In page 7, line 116, a reference is needed for the dehalogenative reduction under the BCF-hydrosilane catalytic system.

Our response: Reference was added to the citation list according the reviewer's valuable suggestion.

22. In page 11, line 188, after the sentence "... (entry 4 in Supplementary Table 1 ...)4." The authors need to discuss about the reversible formation of Int 1 from Int 2, together with a relation to the D-scrambling observed in the experiments of Scheme 6 b and 6c. In this reversible formation of Int 1, BCF works as a Lewis acid to abstract the hydride. A suitable paper(s) "DOI: 10.1039/c9qo01437c" and "Ref. 11" need to be cited for this reversibility.

Our response: Corresponding discussion and references were added to the revised manuscript according to the reviewer's valuable suggestion.

22. Accordingly, in Scheme 7, the arrows are needed between Int1 and Int2 to reflect the reversible formation of Int1 from Int2.

Our response: Changed as suggested.

23. In Scheme 7, "RDS" better to be put on the conversion flow from 5aa (via TS2/3) to Int5 in the Ring Isomerization cycle. In addition, "fast" is good to put on the arrow over from Int5 to TS6/7 to 6aa.

Our response: Changed as suggested.

24. This reviewer suggests to put the ground energies of key species containing "resting intermediate (-14.0)", BCF (0.0 kcal/mol), Int1 (-2.7), Int2 (-11.1), and product (-37.1).

Our response: Changed as suggested.

REVIEWERS' COMMENTS

Reviewer #1 (Remarks to the Author):

The authors appropriately responded to the most of this reviewer's concerns. I suggest that this work can be published in Nature Communications due to the novel sila-benzoazole synthesis, atom-economic procedure, excellent yields, and fully and clearly presented data after few minor revisions below.

- In page 2, line 18, revise "systematically mechanistic studies" -> "systematical mechanistic studies"
- In page 7, line 45, revise "exchange reaction" -> "reduction reaction". The authors may revise the sentence accordingly.

Reviewer #2 (Remarks to the Author):

The authors addressed the issues quite clearly that the reviewer raised in the original manuscript! The current revised text is now in a better shape and is ready to go for publication this reviewer thinks. Again, this work is a good example to underline an importance of finding a new (catalytic) reactivity, which could offer an economic and convenient synthetic route towards a series of unique structures of compounds.

Reviewer 1:

Reviewer's general comments: The authors appropriately responded to the most of this reviewer's concerns. I suggest that this work can be published in Nature Communications due to the novel sila-benzoazole synthesis, atom-economic procedure, excellent yields, and fully and clearly presented data after few minor revisions below.

Our response: We greatly appreciate the reviewer's positive recommendation for publication in *Nature communications*.

Reviewer's specific questions:

1. In page 2, line 18, revise "systematically mechanistic studies" -> "systematical mechanistic studies"

Our response: Changed as suggested.

2. In page 7, line 45, revise "exchange reaction" -> "reduction reaction". The authors may revise the sentence accordingly.

Our response: Changed as suggested.